# Impact of Attachment Style and Temperament Traits on the Quality of Life of Patients with Psoriasis

**DOI:** 10.3390/bs14060434

**Published:** 2024-05-22

**Authors:** Maria Luisa Pistorio, Tania Moretta, Maria Letizia Musumeci, Claudia Russo, Francesco Lacarrubba, Antonino Petralia, Giuseppe Micali, Concetta De Pasquale

**Affiliations:** 1Vascular Surgery and Organ Transplant Unit, Department of General Surgery and Medical-Surgical Specialties, University Hospital of Catania, 95100 Catania, Italy; 2Department of General Psychology, University of Padova, 35131 Padova, Italy; tania.moretta.1@unipd.it; 3Dermatology Clinic, Department of General Surgery and Medical-Surgical Specialties, University Hospital of Catania, 95100 Catania, Italy; marialetizia.musumeci@policlinico.unict.it (M.L.M.); francesco.lacarrubba@unict.it (F.L.); giuseppe.micali@unict.it (G.M.); 4Department of Educational Sciences, University of Catania, 95100 Catania, Italy; claudiarusso95@hotmail.it (C.R.); depasqua@unict.it (C.D.P.); 5Department of Clinical and Experimental Medicine, Institute of Psychiatry, University of Catania, 95100 Catania, Italy; petralia@unict.it

**Keywords:** clinical psychology, attachment, temperament, psoriasis, quality of life, mental health

## Abstract

Background: Psoriasis is a chronic inflammatory skin disease with manifestations that go beyond the visual manifestation, and include psychological aspects. Some mental disorders or personality traits in psoriasis patients have also been highlighted, such as a negative or problematic attitude towards life, impulsive or avoidant behavior, and lower satisfaction with life. The aim of our cross-sectional study was to explore the associations between adult attachment, temperament, and quality of life of patients with psoriasis. Methods: A sample of 75 patients with psoriasis was evaluated with the Attachment Style Questionnaire (ASQ) to study adult attachment, the Temperament Evaluation of Memphis, Pisa, and San Diego Auto-questionnaire (TEMPS-A) to study temperament traits, and the Dermatology Life Quality Index (DLQI) to study the impact of dermatological diseases on patients’ lives. Results: Depressive, cyclothymic, and irritable temperaments were found to be significantly positively associated with a need for approval and preoccupation with relationships subscales of the ASQ. The severity of skin disease effect on the patient’s life was higher in women than in men. Moreover, a statistically significant effect of the need for approval subscale of the ASQ was found. The positive correlation between the severity of skin disease effect on the patient’s life with a need for approval was statistically significant and stronger in women than in men. Conclusions: A better understanding of the impact of mental comorbidities on psoriasis and vice versa places an ever-greater responsibility on dermatologists involved in the management of psoriasis to recognize these problems and collaborate with psychologists and psychiatrists to help these patients.

## 1. Introduction

### 1.1. Relationship between Psychological Factors and Skin Diseases

The skin and the central nervous system are embryologically related and share several hormones, neurotransmitters, and receptors. The skin plays a key role as a sensory organ in socialization processes throughout the entire life cycle. It is reactive to emotional stimuli and its appearance is very important as it influences body image and self-esteem [1,2,3,4].

It is not surprising that a relationship between psychological factors and skin diseases has long been hypothesized. It is common opinion, in fact, that many cases of skin diseases are caused by psychological stress, are linked to certain personality traits, or represent a complication of a mental disorder.

The review by Rousset [5] explored the link between stress and psoriasis. Studies suggest that stress may play a role in triggering the disorder in susceptible individuals. Stress is also one of the consequences of the spread of psoriasis; therefore, understanding its role may prove useful in the treatment of patients with psoriasis. Furthermore, some controlled studies examined in this review have demonstrated the effectiveness of some therapies such as biofeedback and cognitive behavioral therapy.

### 1.2. Dermatological Diseases and Quality of Life

The body of research that looked into the quality of life of people suffering from dermatological diseases proved to be particularly productive. An increase in physical, psychological and social discomfort has emerged in these patients, especially in subjects suffering from psoriasis [6,7,8]. From a study conducted by Mease and Menter [9], quality of life is negatively affected in the case of patients with extensive forms of psoriasis or in cases in which the lesions affect exposed areas of the body such as the hands, face, or delicate areas such as the genitals. Furthermore, variables such as skin pain and skin discomfort [10,11,12] negatively affect some vital functions such as sleep, mood, and, indeed, quality of life [13,14,15]. International research described how the most important comorbidities of psoriasis are mental disorders, especially depression and anxiety [16,17,18,19]. In particular, younger patients and those with a more severe degree of psoriasis are more at risk [20,21,22].

### 1.3. Temperament in Patients with Psoriasis

In the dermatological field, several researchers have attempted to highlight a personality style typical of psoriatic patients [23]. According to the results of the studies by Kilic [24] and Ak [25], psoriatic individuals would present themselves on a temperamental level as inhibited, shy individuals and worried about the potential negative consequences of their behavior. From a character point of view, they have demonstrated a low level of self-acceptance, an attitude that is not very responsible and not very focused on achieving goals.

On the contrary, opposite results were recorded by Karaca [26] and Doruk [27], who did not identify significant differences in terms of temperament and character in patients with psoriasis, vitiligo, and neurodermatitis, either in comparison with each other or with a control group. In light of these results, we can state that the data obtained from studies in this area of research are still controversial and insufficient.

Marek-Józefowicz [28] evaluated the relationship between affective temperament traits and the intensity of depressive symptoms in 208 patients with psoriasis. The results of the study highlighted a specific affective temperament profile in these patients, with higher scores in depressive, anxious and irritable temperament traits.

Pancar Yuksel [29] studied the relationship between temperament profile and perceived stress in patients with psoriasis and found that depressive, cyclothymic and anxious temperaments are significantly associated with a higher level of perceived stress.

Janowski [30] found that the severity of the dermatological disorder and the quality of life of 150 patients with psoriasis were moderated by some temperament traits and that, therefore, temperament can be considered a risk factor.

In the study by Litaiem [31], the temperament profile was studied in 65 patients with psoriasis. The results showed some significant differences in gender: women reported higher scores in depressive and anxious temperaments, while there were no significant differences regarding the temperament scores of healthy controls. As the duration of the disorder increased, irritable temperament scores tended to decrease.

### 1.4. Psychosomatic Skin Diseases and Attachment

To delve deeper into the emotional organization of the psychosomatic patient and the attribution of its meanings, the researchers tried to reconstruct the history of the patient’s development and analyze the mental state relating to attachment with significant figures.

In adulthood, insecure individuals (anxious-ambivalent or avoidant) experience problems identifying and expressing their feelings, especially following stressful social-emotional situations. This is why this type of individual attributes their emotions to the outside world or interprets them as symptoms [32]. From the results of a case–control study conducted by Russiello [33], it emerged that the most dangerous predisposing factor identified in subjects suffering from psychosomatic skin diseases was being an adult with an unresolved attachment to significant figures, which is equivalent to the child’s insecure attachment (avoidant and anxious-ambivalent).

In the study by Esposito [34], the attachment style was studied in 105 patients with psoriasis, and it was found that the worried (anxious) attachment style is a predictor of a higher level of disability and worse quality of life. In these psoriatic patients, a therapeutic intervention targeting dysfunctional attachment may be useful.

Other research conducted in the field of psychosomatic disorders has demonstrated how these patients present the phenomenon of alexithymia in a higher percentage than that observed in controls. The highest prevalence of alexithymia was found in patients with psoriatic arthritis, patients with severe psoriasis, and patients with psoriasis with visible skin lesions [27,35,36,37].

The present study focused on a sample of patients with psoriasis to:Explore the bivariate relation between attachment style, temperament and quality of life.Explore the association between the dependent variable “quality of life” and the independent variables “attachment styles and temperament”. It is hypothesized that a secure attachment style positively influences quality of life, while an insecure attachment style and temperament traits negatively influence quality of life.Explore any gender differences for the variables studied.

## 2. Materials and Methods

### 2.1. Participants and Procedure

In the period between 1 March and 31 May 2022, a survey was conducted on a sample of 75 outpatients suffering from psoriasis at the dermatology unit of the University Hospital of Catania. The initial sample consisted of 80 subjects. During the psychological evaluations, 3 subjects decided to refuse to continue the study for personal reasons and 2 subjects did not show up for the appointment, not even answering phone calls to ask the reasons. The sample examined included patients over the age of 18, without dementia or severe cognitive impairment, who, by appointment, went to the hospital to carry out an initial medical examination or a check-up. After being notified by a dermatologist in the department, who explained the study in question and requested their participation, the patients who had given consent were accompanied to a special room to carry out the questionnaires. The psychological evaluations were carried out by two trainee psychologists, who guaranteed their presence for the entire period of the research.

Generally speaking, the questionnaires were self-administered; however, if the patient showed evident difficulties in reading and understanding the items, the tests were administered by the researcher.

Participants voluntarily enrolled in this cross-sectional study. No compensation was offered for their participation. Prior to their involvement, participants were fully informed about the study’s objectives and procedures, and they provided their written informed consent. This study was approved by the local ethics committee (Comitato Etico “Catania 1” Approval Code: 25940; Approval Date: 2019) and adhered to the ethical guidelines established by the Italian Psychological Association and the principles outlined in the 1964 Declaration of Helsinki, as well as its subsequent revisions [38].

Given that the present study is the first to study the association between attachment styles and temperamental traits with the severity of skin disease effect on the patient’s life, there was no related effect size to choose from for formal power analysis. The present study has been conducted as a first hypothesis testing and should be used to design larger confirmatory studies.

### 2.2. Measures

Among the materials and methods used, after the initial cognitive interview, adult attachment was evaluated by the “Attachment Style Questionnaire” (ASQ) by Feeney, Noller and Hanrahan [39]. It is a self-report questionnaire composed of 40 items that aims to measure the dimensions of attachment and the styles that distinguish it. The interviewee must indicate, for each item, their degree of agreement/disagreement, using a 6-point scale (from 1 = totally disagree to 6 = totally agree). The ASQ is made up of five subscales that identify the main dimensions of attachment: “Trust”, “Discomfort with intimacy”, “Secondariness of relationships”, “Need for approval”, and “Concern for relationships”. According to the authors, the ASQ is able to effectively discriminate between secure and insecure attachment. The latest Italian version of the ASQ, translated by Fossati and collaborators [40,41], was used in the research and administered to a sample of psychiatric patients and to a non-clinical sample. It is suggested to use the 75th or 90th percentile of the individual scales for Insecure Attachment, and the 25th or 10th percentile for Trust [41].

To evaluate temperament traits, the TEMPS-A “Temperament Evaluation of Memphis, Pisa and San Diego Auto-questionnaire” was used, which is a semi-structured interview in the form of a self-administered questionnaire, designed by Akiskal and his collaborators in the Memphis research group -Pisa-Paris, San Diego [42,43], which was validated in an Italian population of 1.010 students aged between 14 and 25 years old [44]. In this research, the version of Maurizio Pompili and Paolo Girardi et al. was used [45], taken from the study “TEMPS-A (Rome): Psychometric validation of affective temperaments in clinically well subjects in central and southern Italy”. This instrument is composed of 110 items (109 in the male version), with a dichotomous yes–no response, necessary to quantify and determine the affective temperament in reference to the patient’s entire life. The TEMPS-A includes, in each temperament scale, sections such as “emotional reactivity” (i.e., depressive, labile, irritable, joyful), “cognitive” (i.e., pessimism vs. optimism), “psychomotor” (i.e., low vs. high energy), “circadian” (related to sleep) and “social” (i.e., some behavioral traits of a follower and/or leader, frequent falling in love and/or breaking up of romantic relationships). The tool takes into account five dimensions: depressive, cyclothymic, hyperthymic, irritable, and anxious. The questions relating to these temperament traits were all grouped together. As regards depressive, cyclothymic and hyperthymic temperament, it is possible to obtain a maximum score of 21 points for each of these domains; with respect to irritable temperament, a maximum of 21 points are obtained in the female version and 20 in the male version. For anxious temperament, it is possible to reach a maximum score of 26 points. After adding the scores obtained in the various subscales, the result is divided by the total number of questions in order to calculate the points for each temperament. The TEMPS-A measures the severity of temperament traits from 0 to 1. Affective temperaments were determined according to their z-scores in each subscale. An affective temperament was considered to be manifested as “dominant” if its z-score was at least 2 standard deviations (SD) above the z-score of the normative sample [46]. The reliability of TEMPS-A was assessed by Cronbach’s alpha coefficients for the components and they were quite high; the alpha computed for the first subscale, with the largest number of items, was 0.89, for the Irritable, it was 0.77, and for the Hyperthymic, it was 0.74 [46].

Finally, the study patients completed the Dermatology Life Quality Index (DLQI) by Finlay [47], which is one of the most used tools to evaluate the impact of dermatological diseases on patients’ lives. This is a self-administered questionnaire developed to measure the impact of skin conditions on a patient’s life in the previous week. It contains 10 questions, grouped into six subscales: symptoms and feelings (items 1 and 2), daily activities (items 3 and 4), free time (items 5 and 6), work and school (items 7), personal relationships (items 8 and 9), and treatment (item 10). For each item, the participants were asked to assign a score of “0” for “not at all” or “not relevant”, “1” for “a little”, “2” for “a lot”, and “3” for “very, very much”. Item scores are added to obtain a maximum score of 30. Higher scores indicate greater impairment in quality of life. In this study, the DLQI score was classified as 0–1 (no affected), 2–5 (mild affected), 6–10 (moderate affected), and >10 (severe affected), and a severe affected life quality among psoriasis patients was identified when the total DLQI score equaled or exceeded 10 [48]. This study used the validated Italian version of the DLQI, Dermatology Life Quality Index. The internal consistency of the DLQI was high (Cronbach’s alpha of 50.83) [49].

### 2.3. Statistical Analysis

All analyses were performed using R software, version 4.0.2 [50]. Two participants were omitted from the analysis because they provided incomplete responses to the questions.

Pearson’s correlation coefficients were used to assess the strengths of relationships among the study variables (see Table 1).

Multiple regression analyses were used to study the relative statistical power of the associations of (i) demographic characteristics (i.e., sex and age), (ii) attachment styles (i.e., confidence [secure attachment], need for approval and preoccupation with relationships [anxious/ambivalent style], discomfort with closeness and relationships as secondary [avoidant style], as identified by the ASQ subscales), and (iii) temperamental traits (i.e., depressive, cyclothymic, hyperthymic, irritable, and anxious, as identified by the TEMPS-A subscales) with the severity of skin disease effect on the patient’s life (identified by the DLQI). Specifically, considering the severity of skin disease effect on the patient’s life as the dependent variable, linear equations were defined by starting from a simple equation including the participants’ sex and age at baseline (Model 1) and adding attachment styles and temperamental traits to the subsequent models (Model 2 and Model 3). The adjusted squared multiple correlation coefficient (R2) was used to determine the amount of variation explained in the skin disease effect on the patient’s life. The Akaike Information Criterion (AIC) and the Bayesian Information Criterion (BIC) were used to select the model that more appropriately described the data; that is, the model with the smallest AIC and BIC was considered to have the best fit to the dependent variable.

Multicollinearity was monitored by examining the variance inflation factor (VIF). In the present study, the VIF measure indicated that multicollinearity was not a concern (all VIFs ≤ 3.44). Moreover, before running the multiple regression analysis, the data were examined for skewness, kurtosis, outliers, and normalcy (in this study, all the values of skewness were between −0.36 and 1.56, and the values of kurtosis were between −1.26 and 2.20). The scatterplot of the standardized residuals showed that the data met the assumptions of homogeneity of variance and linearity.

The maximum likelihood method was used to analyze the contribution of parameter variables within the selected model, and the differences in the AIC and BIC of the model with and the model without the parameters (ΔAIC and ΔBIC) were estimated to quantify the strengths of the parameter evidence [51].

Finally, potential sex differences between associations of statistically significant predictors and the dependent variable were tested by transforming the correlations into z-scores using Fisher’s r-to-z transformation [52].

## 3. Results

Descriptive statistics and Pearson’s correlation coefficients are reported in Table 1. In addition to the expected correlations between subscales of the ASQ and the TEMPS-A, the skin disease effect on the DLQI scores was significantly negatively associated with the confidence subscale of the ASQ and hyperthymic temperament. Moreover, it was positively associated with discomfort with closeness, need for approval, and preoccupation with relationships subscales of the ASQ, as well as cyclothymic and anxious temperaments. Furthermore, depressive, cyclothymic, irritable, and anxious temperaments were found to be significantly positively associated with discomfort with closeness, need for approval, and preoccupation with relationships subscales of the ASQ, and negatively associated with the confidence subscale of the ASQ. Conversely, the hyperthymic temperament was significantly negatively associated with discomfort, with closeness and preoccupation with relationships subscales of the ASQ, and positively associated with the confidence subscale of the ASQ.

Table 2 shows the R2, BICs, and AICs of fitted multiple regression models (i.e., Model 1, Model 2, and Model 3). As indicated by the model selection method on the severity of skin disease effect on the patient’s life, the regression model that considered both demographic information (i.e., sex and age) and attachment styles (i.e., confidence, need for approval and preoccupation with relationships, discomfort with closeness and relationships as secondary) as predictors fitted the data the best (Table 2). The effect of predictors was tested by the maximum likelihood method. A statistically significant effect of sex was found (F = 11.34, *p* = 0.001, ΔBIC = −7.45; ΔAIC = −9.74; Figure 1), indicating that the severity of skin disease effect on the patient’s life was higher in women (M = 4.41, SD = 4.53) than in men (M = 1.47, SD = 2.61). Moreover, a statistically significant effect of the need for approval subscale of the ASQ was found (β = 0.14, 95%CI [0.01 0.28], *p* = 0.005, ΔBIC = −0.41, ΔAIC = −2.70; Figure 2); the higher the need for approval, the higher the severity of skin disease effect on the patient’s life. Lastly, a statistically significant small effect of the confidence subscale of the ASQ was also found (β = −0.04, 95%CI [−0.21 0.12], *p* = 0.005); however, this effect showed low strength of evidence, as indicated by ΔBICand ΔAIC (1.71).

To test, in an exploratory fashion, for potential differences between the relationship of the severity of skin disease effect on the patient’s life with the need for approval subscale of the ASQ in women (r = 0.55, *p*  <  0.001) and men (r = 0.12, *p*  =  0.49), Pearson’s coefficients were transformed into z-scores using Fisher’s r-to-z transformation. The positive correlation between the severity of skin disease effect on the patient’s life with the need for approval was statistically significant and stronger in women than in men (z  =  2.03, *p*  =  0.02).

## 4. Discussion

### 4.1. Psoriasis and Mental Health

Psoriasis is a chronic inflammatory skin disease with manifestations that go beyond the visual manifestation, and include psychological aspects. Some personality disorders or personality traits in psoriasis patients have also been highlighted, such as a negative or problematic attitude towards life, impulsive or avoidant behavior, and lower satisfaction with life [53,54,55]. Moreover, the association between psoriasis and mental health disorders is now widely accepted, especially depression and anxiety [56,57,58,59].

Psoriasis causes discomfort that affects the emotional sphere, interpersonal relationships, and daily activities of patients [60,61,62]. In our study, it emerged that the quality of life of patients with psoriasis is negatively associated with the secure attachment style and with the hyperthymic temperament trait, while it is positively associated with the insecure attachment styles (discomfort with intimacy, need for approval and concern about relationships). Insecure attachment styles (discomfort with intimacy, need for approval, and concern for relationships) were positively associated with the depressive, cyclothymic, irritable, and anxious temperaments.

### 4.2. Psychological Impact of Psoriasis in Women

Manifesting itself on the skin, psoriasis certainly has strong psychological repercussions on females: data showed that women experience the social stigma of the disease to a greater extent and that they often somatize it more than males, with a higher risk of stress, anxiety and depression, and a negative impact on their overall sense of satisfaction. This can lead to a greater risk of loneliness and isolation [63,64,65].

In our study, the regression model, with sex, age and attachment styles as predictors, highlighted that the dermatological quality of life is more compromised in women than in men. Furthermore, the need for approval appears to have a statistically significant effect on a worse quality of life, particularly for women compared to men.

The need for approval reflects the need for acceptance and confirmation from other people and corresponds to fearful and preoccupied attachment styles. This indicates fear of intimacy: fearful subjects have a negative view of both themselves and others; they desire social contact and intimacy, but they do not trust others and fear rejection; thus, they avoid social situations.

In women, this pathology can have a very particular physical and psychological impact. Many women wonder whether the disease could affect a pregnancy and reproductive health, while many others fear that it could affect their physical appearance and their relationship as a couple [66,67].

In women, the progress of the pathology can vary depending on their hormonal phase: studies have shown how taking oral contraceptives with high doses of estrogen is associated with an improvement in psoriasis, while worsening is recorded in the days preceding the menstrual cycle, during menopause and immediately after giving birth [68,69,70,71,72].

### 4.3. Psychological Interventions of Psoriasis

Thanks to numerous studies that identify factors (psychological symptoms, global distress, impaired quality of life) that are important for psoriasis, patients can receive the appropriate psychological and social support [73,74,75].

Relaxation techniques, cognitive behavioral therapy, and support groups have a positive effect on the treatment of psoriasis: they reduce the patient’s stress level, promote emotional control and adequate self-esteem, which lead to acceptance of the disease and improve the patient’s quality of life [2,76,77]. It seems clear that a better understanding of the impact of mental comorbidities on psoriasis places an ever-greater responsibility on dermatologists involved in the management of psoriasis in collaborating with psychologists and psychiatrists to help these patients. Early diagnosis and treatment are very important. Several studies, moreover, demonstrate that psoriatic women have higher expectations of therapies than men; in fact, they expect these to allow them to return to being active so as not to create too many repercussions on the family [78].

Some limitations of our study have to be considered: the small sample size, the cross-sectional nature of the design, the use of self-report measures, and the lack of a control group. However, the results of our study carry significant clinical implications. Indeed, the findings of our research underscore the importance of giving increased attention to adult attachment, temperamental traits, and quality of life during the assessment of patients with psoriasis, as these factors may serve as protective or risk factors for these patients. Identifying these variables in a timely manner will help to provide targeted psychotherapeutic interventions to enhance the acceptance and management of the disease in patients with psoriasis, especially women.

Therefore, we recommend screening for aspects of psychopathology of these patients and the collaboration of dermatologists with psychologists and psychiatrists to ensure psychological and/or pharmacological support where necessary to ensure the psychological well-being of dermatological patients [79,80,81,82].

## 5. Conclusions

Beyond the physical dimensions of the disease, psoriasis has a broad emotional and psychosocial effect on patients, affecting social functioning and interpersonal relationships. The recognition of psoriasis, as well as its associated psychological and psychiatric comorbidities, may facilitate appropriate management of this disorder, with targeted therapies, such as psychological interventions, psychotherapies and psychopharmacological interventions, as needed, and positively affect patient outcomes. Future research should examine the temperament and attachment of patients with psoriasis, as well as other dermatological diseases, so that targeted treatments can be provided and the quality of life of these patients improved.

## Figures and Tables

**Figure 1 behavsci-14-00434-f001:**
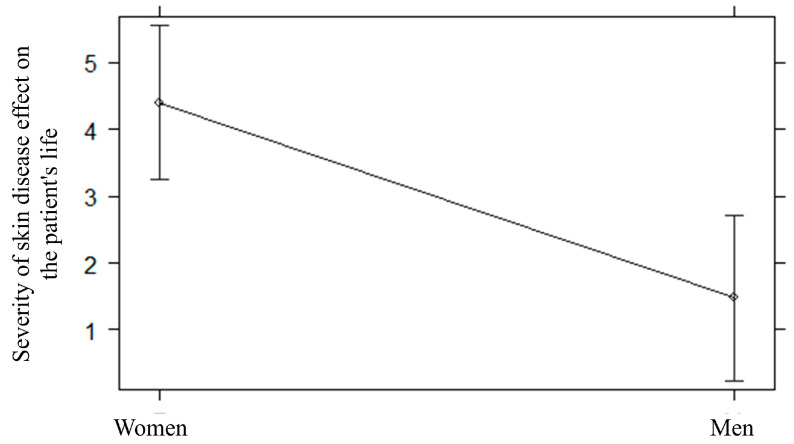
Sex differences in the severity of skin disease effect on the patient’s life.

**Figure 2 behavsci-14-00434-f002:**
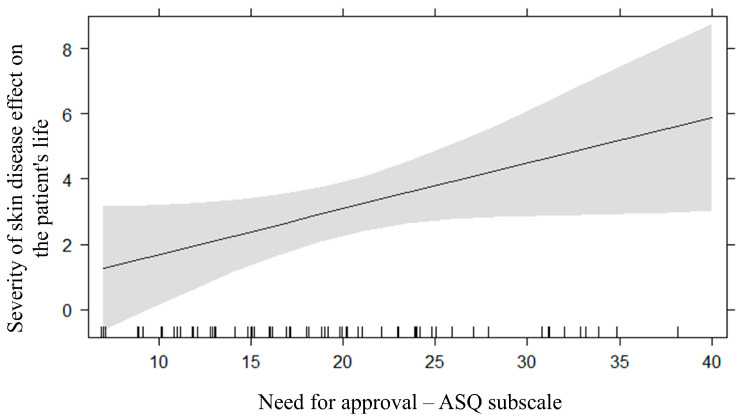
The relationship between the need for approval subscale of the attachment style questionnaire (ASQ) and the severity of skin disease effect on the patient’s life.

**Table 1 behavsci-14-00434-t001:** Descriptive statistics and intercorrelations between main study variables.

n = 73	n/Mean ± sd	Pearson’s Correlation Coefficients										
Sex (F/M)	39/34	1.	2.	3.	4.	5.	6.	7.	8.	9.	10.	11.
1. Age	47.79 ± 15.50	1										
2. ASQ Confidence	40.47 ± 6.12	0.06	1									
3. ASQ Discomfort with closeness	45.27 ± 10.54	0.01	−0.48 ***	1								
4. ASQ Need for approval	19.66 ± 8.12	−0.01	−0.38 **	0.39 **	1							
5. ASQ Preoccupation with relationships	30.99 ± 8.72	−0.02	−0.36 **	0.48 ***	0.61 ***	1						
6. ASQ Secondary with relationships	17.26 ± 7.41	−0.05	−0.01	0.38 **	0.31 *	0.21	1					
7. TEMPS-A depressive	6.97 ± 3.50	0.20	−0.32 *	0.51 ***	0.51 ***	0.56 ***	0.19	1				
8. TEMPS-A cyclothymic	5.19 ± 4.22	−0.02	−0.32 *	0.42 ***	0.45 ***	0.53 ***	0.16	0.67 ***	1			
9. TEMPS-A hyperthymic	8.90 ± 4.27	0.04	0.56 ***	−0.25 *	−0.21	−0.25 *	−0.06	−0.05	0	1		
10. TEMPS-A irritable	2.25 ± 2.53	0.04	−0.31 *	0.38 **	0.32 *	0.49 ***	0.16	0.62 ***	0.73 ***	−0.05	1	
11. TEMPS-A anxious	6.47 ± 4.58	0.12	−0.29 *	0.41 ***	0.33 *	0.30 *	0.14	0.57 ***	0.66 ***	−0.06	0.52 ***	1
12. Dermatology Life Quality Index	3.04 ± 4.02	−0.16	−0.24 *	0.28 *	0.40 ***	0.31 *	0.15	0.20	0.25 *	−0.29 *	0.23	0.34 **

* represents *p* value < 0.05; ** represents *p* value < 0.01; *** represents *p* value < 0.001.

**Table 2 behavsci-14-00434-t002:** Multiple regression models’ comparison by the adjusted squared multiple correlation coefficient (R^2^), the Bayesian Information Criterion (BIC), and the Akaike Information Criterion (AIC). Dependent variable: severity of skin disease effect on the patient’s life.

Models	Predictor(s)	R^2^	BIC	AIC
Model 0	~1		417.82	413.24
Model 1	~Sex + Age	0.14	413.24	404.08
Model 2	~[Model 1 predictors] + ASQ Confidence + ASQ Discomfort with closeness + ASQ Need for approval + ASQ Preoccupation with relationships + ASQ Secondary with relationships	0.25	419.31	398.70
Model 3	~[Model 2 predictors] + TEMPS-A depressive + TEMPS-A cyclothymic + TEMPS-A hyperthymic + TEMPS-A irritable + TEMPS-A anxious	0.26	433.88	401.81

## Data Availability

The data that support the findings of this study are available on request from the corresponding author. The data are not publicly available due to privacy or ethical restrictions.

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
