# Peer review of "Impact of Attachment Style and Temperament Traits on the Quality of Life of Patients with Psoriasis"

_behavsci, 2024, doi:10.3390/bs14060434_

Round 1

Reviewer 1 Report

Comments and Suggestions for Authors

The work examines the associations between adult attachment, temperament and quality of life life of patients with psoriasis.
The summary has been prepared correctly, the introduction of the work ends with clearly defined goals.
Is the literature citation written in accordance with the journal's requirements? The authors provide the sequential citation number and the year of publication of the literature item.
The methodological part was developed properly. Why did the authors use the Pearson correlation test and not the Spearman correlation test? There is also no information about the distribution of variables, whether they were parametric or non-parametric variables.
Unfortunately, I did not find any tables or figures in the reviewed work or in the additional materials.
The discussion discusses the results with reference to data in the literature. The work ends with conclusions that meet the purpose of the work.

Reviewer 2 Report

Comments and Suggestions for Authors

Reviewer Report

This study aims to clarify the relationship between psoriasis and adult attachment, temperament, and quality of life in patients with psoriasis.

The findings of this study underscore the crucial need to deepen our understanding of the reciprocal impact of psychiatric comorbidities and psoriasis. This insight is not only academically significant but also has profound implications for patient care. It underscores the necessity for dermatologists, psychiatrists, and psychologists to collaborate in supporting patients with psoriasis. This research paper, therefore, serves as a cornerstone in establishing a comprehensive methodology for the treatment of psoriasis patients.

In the research methodology, I affirm that the measurement methods for data acquisition, statistical analysis, and other aspects were meticulously planned and executed. This rigorous approach ensures the reliability and validity of the study's findings, enhancing its credibility.

However, it should be emphasized that no specific data indicating the research's results are presented. This paper lacks data on psoriasis patients participating in the study, which should be presented in Table 1, and multiple regression models, which should be presented in Table 2. The data can be read from the text, but the publication in tabular form should be double-checked.

Major Comments

This study states that 75 subjects participated in the analysis. Clarifying the changes in the number of research collaborators or analysis subjects over time, from the beginning of the study to the end of the analysis, is important for the accuracy of the study. Therefore, it should be clearly stated on what basis the number of research collaborators was set. If there were withdrawals from participation in the study after recruitment, the number of people and the reasons should be clearly described.

For all chronic diseases, we believe that the number of years since disease onset and duration are critical factors, in addition to information such as gender and age of disease patients. In this study, you should indicate what kind of thinking you used to analyze the number of years since disease onset. Please also check again regarding the multiple regression model. Although I have not been able to check Tables 1 and 2, I have not been able to check them accurately.

In the DISCUSSION, I confirm that the considerations from the study results are well documented. In addition to these considerations, the purposefulness of this study regarding psoriasis disease would be further clarified by comparing the results with those of diseases other than psoriasis disease, for which the psychological relationship has been clarified. If possible, please consider publishing comparative studies with other diseases.

Reviewer 3 Report

Comments and Suggestions for Authors

It is worthwhile to explore psychosocial variables that are related to quality of life in patients with psoriasis. Overall, no major problems appear in the conduct of the study. However, improvement is needed in writing the paper. What I think would like to be improved is as follows.

1. You can omit statistical values from the abstract unless there is a special need to present them.

2. Please check the citation style of this journal.

3. Please structure your paragraphs in the introduction and discussion to improve readability. There are connections between sentences or more explanation is needed.

4. For example, the numbering style for Research Objectives is the same as that for Introduction, Methods, and Results.

5. Please provide clear explanations of measurement tools. Provide information on the reliability and validity of the measurement instrument. At least internal consistency should be presented.

6. When you present statistical values, the average and standard deviation of the variables themselves are not that important, so it is better to present them only in tables and omit them from the text. This is because readers may become fixated on the values themselves.

7. r is already promised to be Pearson’s r. So just mark it with r.

Reviewer 4 Report

Comments and Suggestions for Authors

I would like to thank the editors and authors for the opportunity to review the article "Impact of attachment style and temperament traits on the quality of life of patients with psoriasis"

The chosen topic is of great importance. The chosen topic is of great importance. Psoriasis goes far beyond its physical manifestations, exerting a profound emotional and psychosocial impact on patients, interfering with their social interactions and interpersonal relationships. In this context, the present study reveals results of significant clinical relevance, highlighting the importance of identifying variables such as emotional attachment, temperamental traits, and quality of life. These aspects can play a crucial role in the development of therapeutic strategies aimed at promoting acceptance and effective management of the disease in patients with psoriasis.

In general terms, I can say that although the article presents specific references of interest, only a small percentage of the references cited in the article are recent, with only 31% being less than six years old. To ensure the currency and relevance of the cited literature, I recommend including more recent references that contribute to the understanding of the topic.

As I will comment below, I have the impression that the manuscript needs a major revision and clarification of some aspects.

I will now offer my contributions or suggestions for improving the manuscript:

Abstract

Specify the study design (the study design only appears in the limitations section).

Introduction

The introduction needs to be revised with the inclusion of recent references, providing readers with a current scientific foundation and justification for the study.

Materials and methods

Describe the type of study, as well as the eligibility criteria, and the sources and selection of participants, inclusion and exclusion criteria, and procedures.

Clarify how the anonymity, confidentiality and privacy of the subjects were safeguarded. Authors must clarify who had access to the collected data, how they were protected and the expected destruction.

Explain how the study size was arrived at and explain how missing data were addressed.

It's essential to offer a clear description of the scales used, including whether established cutoff values exist. These cutoff values are crucial for interpreting study findings and identifying different risk or severity levels among individuals.

It is suggested to include in the article the approval from the ethics committee, along with the corresponding approval number.

Results

The authors are requested to clarify the number of participants. In the method, it is mentioned that there are 75 participants, but in Table 1, the authors show that there are 39 female participants and 34 male participants, totaling 73 participants. In other words, there is a difference of 2 participants. It is suggested to clarify the loss of 2 participants.

Figures 1 and 2 do not have titles; it is suggested that they be included.

Discussion

It's essential to update references with the most recent literature, preferably from the last five years, to ensure the inclusion of the latest findings and advancements in the field. Integrating recent references will strengthen the discussion and provide readers with up-to-date information on the topic. By implementing these changes, the article will become more reader-friendly and academically rigorous, enhancing its impact and contribution to the existing body of knowledge.

Conclusions

Some conclusions appear in the discussion section; it is suggested that they add a separate chapter for the conclusions and include suggestions for future research.

References

In general terms, I can say that the article presents specific references in interest, however, the majority are over 6 years old (more than 68%).

It is suggested that a review of the references be carried out in accordance with the journal's guidelines, as well as the citations throughout the text.

Final decision:

The manuscript needs several important changes.

I hope that my contributions will serve to improve this article and the study you propose.

Round 2

Reviewer 1 Report

Comments and Suggestions for Authors

The authors made suggested corrections to the manuscript.

Author Response

Thank you for your revision.

Reviewer 4 Report

Comments and Suggestions for Authors

I would like to express my sincere gratitude to the editors and authors for the opportunity to review the article "Impact of attachment style and temperament traits on the quality of life of patients with psoriasis." Overall, I can confirm that the authors have revised the manuscript in response to my suggestions for improvement.

The authors have made improvements to the work. They updated the bibliographic references and enhanced the abstract, making it clearer and more concise. In the Materials and Methods section, they described the scales in more detail, which facilitates understanding. Additionally, they detailed the type of study, the sources and selection of participants, inclusion criteria, procedures adopted, and clarified the loss of two participants. They also included approval from the ethics committee. Furthermore, they added the main conclusions, addressing the proposed objectives, and provided suggestions for future research. However, I noted the inclusion of citations in the conclusion chapter, which should be removed, as conclusions should only reflect the work carried out by the authors themselves, without references to others.

I highlight the significant effort and dedication of the authors to improve the manuscript. Therefore, my current proposal is to accept it for publication with minor revisions for publication (removing the references in the conclusion).

Finally, I would like to congratulate the authors on the work done and their professional conduct throughout the process.

Sincerely,

Author Response

Thank you for your suggestion. We removed citations from the conclusion chapter. Thanks again for your revision.